# Developing a Machine-Learning ‘Smart’ PCR Thermocycler, Part 1: Construction of a Theoretical Framework

**DOI:** 10.3390/genes15091196

**Published:** 2024-09-11

**Authors:** Caitlin McDonald, Duncan Taylor, Gershom Mwachari Masawi, Ayesha Khalid Ahmed Khan, Richard Leibbrandt, Adrian Linacre, Russell S. A. Brinkworth

**Affiliations:** 1College of Science and Engineering, Flinders University, GPO Box 2100, Adelaide, SA 5001, Australia; caitlin.mcdonald@flinders.edu.au (C.M.); masa0033@flinders.edu.au (G.M.M.); khan0302@flinders.edu.au (A.K.A.K.); richard.leibbrandt@flinders.edu.au (R.L.); adrian.linacre@flinders.edu.au (A.L.); russell.brinkworth@flinders.edu.au (R.S.A.B.); 2Forensic Science SA, GPO Box 2790, Adelaide, SA 5001, Australia

**Keywords:** PCR thermocycler, cycling conditions, machine learning, STR DNA profile

## Abstract

The use of PCR is widespread in biological fields. Some fields, such as forensic biology, push PCR to its limits as DNA profiling may be required in short timeframes, may be produced from minute amounts of starting material, and may be required to perform in the presence of inhibitory compounds. Due to the extreme high-throughput of samples using PCR in forensic science, any small improvement in the ability of PCR to address these challenges can have dramatic effects for the community. At least part of the improvement in PCR performance could potentially come by altering PCR cycling conditions. These alterations could be general, in that they are applied to all samples, or they could be tailored to individual samples for maximum targeted effect. Further to this, there may be the ability to respond in real time to the conditions of PCR for a sample and make cycling parameters change on the fly. Such a goal would require both a means to track the conditions of the PCR in real time, and the knowledge of how cycling parameters should be altered, given the current conditions. In Part 1 of our work, we carry out the theoretical groundwork for the ambitious goal of creating a smart PCR system that can respond appropriately to features within individual samples in real time. We approach this task using an open qPCR instrument to provide real-time feedback and machine learning to identify what a successful PCR ‘looks like’ at different stages of the process. We describe the fundamental steps to set up a real-time feedback system, devise a method of controlling PCR cycling conditions from cycle to cycle, and to develop a system of defining PCR goals, scoring the performance of the system towards achieving those goals. We then present three proof-of-concept studies that prove the feasibility of this overall method. In a later Part 2 of our work, we demonstrate the performance of the theory outlined in this paper on a large-scale PCR cycling condition alteration experiment. The aim is to utilise machine learning so that throughout the process of PCR automatic adjustments can be made to best alter cycling conditions towards a user-defined goal. The realisation of smart PCR systems will have large-scale ramifications for biological fields that utilise PCR.

## 1. Introduction

The generation of short tandem repeat (STR) DNA profiles has been the dominant forensic biology tool used in forensic laboratories since the 1990s [1]. The modern process of deoxyribonucleic acid (DNA) profiling evolved out of what is considered the first forensic DNA testing carried out by Sir Alec Jeffreys in 1985 [2]. The polymerase chain reaction (PCR) copies regions of interest along the genome, so that small amounts of starting material can be visualised in the form of a DNA profile. PCR works by subjecting a DNA sample (combined with PCR reagents) through a series of cycles (or stages). Each of these cycles include a denaturation step where the DNA is heated to separate the double-stranded DNA (dsDNA) into single-stranded DNA (ssDNA), an annealing step where primers are able to bind to target areas on the ssDNA, and an extension step where the target regions are copied by an enzyme called DNA polymerase [3,4,5,6].

A single PCR program will run through these three steps for 26 to 34 cycles (depending on the program). Different PCR programs may vary the temperatures or times used but, for most PCR programs these three (or two) steps have both a fixed temperature and duration at which the solution is held for each stage of the cycles within a program [1,7]. This has been the case since the inception of PCR in the 1980s [8]. There is one notable extension to this, known as ‘touchdown’ PCR [9], which varies the annealing temperature across the program, i.e., starting at one temperature during cycle 1 and decreasing this temperature in each cycle. The purpose of touchdown PCR is to ensure maximum stringency in the initial cycles, by having the annealing temperature at a level which only perfectly designed (i.e., complementary) primers will anneal. As the temperature of annealing is reduced (lowering the stringency), preferential amplification of short PCR amplicons is hypothesised to exist in the reaction vessel compared to the original template DNA. This type of reaction is particularly useful when primer pairs are multiplexed but have different optimal annealing temperatures [10]. Despite the success it has seen in other disciplines, touchdown PCR has not yet been integrated in forensic applications [7]. The commercialisation of PCR kits in forensic science, and their extensive validation, means most laboratories follow the cycling conditions recommended by the PCR kit manufacturers to ensure amplification is optimal and their results are reproducible and standardised [7,11,12]. 

While the PCR process has been tried and proven over the decades [13], there are likely ways in which the process can be optimised, either being made more effective, more efficient or both. From a conceptual point of view, the composition of the reaction mixture at the first cycle is very different from cycles near the end of the program [7]. As the program progresses, the enzyme used to extend the primed DNA will have lost some efficiency [14]. Also, the reaction will consist of many (millions or billions) of copied fragments near the end of the reaction, as opposed to predominantly being template DNA at the start. We hypothesise that the optimal PCR conditions for these two states of the reaction are different.

The ability to alter PCR cycling conditions is possible with all modern PCR thermocyclers, which allow programming of the cycling conditions for any desired PCR program; however, there are limitations to the configurability of the programs. For example, while a temperature and time can be set for steps in a PCR cycle, there may be no ability to program in gradient changes across the cycles. Furthermore, cycling parameters must be programmed before the process begins, and cannot be altered while the cycling is in progress. This means it is not possible to adapt the parameters, should additional information about the process (such as the current concentration of amplified DNA) become available. These limits on configurability can be addressed by the rise in popularity of do-it-yourself (DIY) biotechnology. Several vendors provide kits, preassembled instruments, or instructions for PCR, or real-time PCR thermocycler instruments (such as the one used in our study). A real-time PCR, also known as quantitative PCR (or qPCR), is a thermocycling instrument with reaction vessel fluorescence monitoring capability. The signal from changes in amplified DNA is collected and monitored throughout the PCR program and typically fed back to a system possessing an observable output (such as a PC) in real time. The fluorescent feedback is ultimately used to quantify the amount of amplified DNA in samples. This information is then typically used to optimally set up amplification reactions that target specific areas of interest for sequencing or fragment sizing.

Other groups have published instructions for printing and milling PCR instruments from base starting materials [15]. As well as being open-source in terms of the hardware, they provide open-source software for complete configurability of cycling conditions. There has also been substantial success integrating biological processes and intelligent computer systems together in the microfluidics sphere [16,17]. Electrowetting-on-dielectric (EWOD) digital microfluidic platforms utilise optical sensors and fluorescence feedback generated during biomolecular interactions (such as PCR) to monitor the progress of reactions in real time [18,19]. In recent years, these integrated platforms have become ideal for PCR assays, due to their high amplification efficiencies [20,21,22], high sensitivity [22,23,24] and low costs [22,23,24]. Furthermore, high-level biological programming languages have been developed that allow for EWOD platforms to be controlled via feedback loops [24,25]. The success of microfluidic PCR assays highlights the value of integrating intelligent computer systems into traditional biochemical processes and the opportunity to further optimise them [7]. Thus, the potential exists for such a method to be used to improve DNA profiling by monitoring the PCR amplification efficiency in real time and using an intelligent computer system. 

One method for improving the effectiveness of PCR is trialling the alteration of temperatures, speed between stages (ramps) and duration of time at each stage in the cycle. This would be under a series of controlled changes and by observing the effect on the resulting amplicons. Given the likelihood that the optimal parameters will be different, depending on the overall goal (e.g., improved speed, quality or quantity of resulting product), and are likely different at different cycles of the PCR process, the parameter space for this problem is very large. However, there is an opportunity to apply machine learning for changes to be learned that will increase the effectiveness of the PCR. The ultimate goal of such a system would be for each sample to have undergone its own bespoke PCR program, perfectly optimised to achieve the best possible results for that sample, with the user able to define what constitutes ‘best’ for their given situation. This would be achieved by carrying out an initial default PCR cycle, then reading some real-time feedback from the PCR vessel, passing that to an algorithm, which will then suggest the optimal PCR cycling conditions for the next cycle based on the overall system goal, and so on, until the program completes. The algorithm will know what cycling parameters to use as a model have been constructed, based on previous PCR cycling data.

There are numerous challenges that must be overcome along the way, before achieving this ultimate goal. In broad terms, these challenges are the following:(1)Defining a goal for PCR ‘effectiveness’.(2)Accessing and controlling a PCR thermocycler in a real-time interactive manner.(3)Carrying out a PCR in a way that provides reliable measurable real-time feedback for an algorithm to use.(4)Creating a metric that represents the quality of the profile.(5)Weighing profile quality against the goal.(6)Carrying out machine learning on a set of training data to inform an algorithm for use in the smart PCR system.

We consider each of these in turn, and describe the work undertaken to address them. Our goal with this work is to develop the framework by addressing the points above, so that machine-learning driven alterations of PCR cycling conditions can begin to be explored.

## 2. Theory

### 2.1. Defining a Goal for PCR ‘Effectiveness’

There are different ways in which the effectiveness of a PCR could be considered. These align directly with goals that a machine-learning algorithm would be trained against.

Perhaps the most impactful goal would be to improve the quality of the DNA profiles produced from extracts. Even then, there are many ways in which the quality of a DNA profile could be considered:peaks within an optimal height range (measured in relative fluorescence units, rfu);More complete DNA profiles;Better balance between peaks within a locus;Better balance between peaks at different loci;Fewer artefacts being generated;Less prominent artefacts, such as lower, or fewer stutter peaks.

Ultimately, these aspects of quality will need to be measurable, balanced against each other for relative importance (e.g., whether a greater peak imbalance be tolerated if it meant more complete DNA profiles were obtained) and distilled into a single score for optimisation. We discuss the details of this process of scoring in Section 2.4. While a general improvement in DNA profile quality is the goal, there may be an opportunity for profile quality improvement in specific situations. For example, PCR that has been carried out in the presence of an inhibitor will classically lead to imbalance between loci (as some loci are more robust to the presence of an inhibitor than others [26]). There may be PCR cycling conditions that could be learnt that are specifically designed to overcome this inhibitory manifestation. As inhibitors are also known to alter the qPCR fluorescence results [27], there are prospects of identifying inhibition in real time, early in the PCR program, and adjusting the proceeding cycling conditions to minimise inhibitory effects.

Another measure of effectiveness could be the time taken for a PCR program to complete. There is a rise in the popularity of ‘rapid’ and portable DNA profiling instruments, which can be taken into the field and used to obtain immediate intelligence [28]. Typically, these instruments carry out a fully automated extraction, PCR, and electrophoresis within 90 min. For these types of instruments, a decrease in PCR time would be a great benefit. A measure of profile quality is still required, as a machine-learning ‘goal’ would be to reduce PCR program time with a minimum (or acceptable) level of degradation in DNA profile quality. This is essential, since without a minimum quality target the machine-learning algorithm will recommend setting the time for each cycle to 0 s, as this reaches the end of the process in the minimum possible time, albeit without any DNA amplification. We speak about how much the goal is weighted towards speed vs. quality in Section 2.5. A separate consideration for speed could be a PCR which recognises early that there will be little-to-no peak information within the profile and so can indicate this without requiring the full PCR program and then capillary electrophoresis to uncover. Conversely, if an acceptable quantity of DNA is present, the machine-learning algorithm could recommend termination of the process before the typical number of cycles are undertaken, since the desired result has already been achieved. In either instance (no detectable, or already sufficient, viable product), recommending aborting the remaining cycles will speed up the process of reaching the inevitable result.

A third goal could be to improve the efficiency of PCR. This could be achieved by either reducing PCR volume or maintaining volume but decreasing the cost (either via amount or type) of reagents used. The purpose here is to maximise the number of PCRs available in commercially provided PCR systems for a given budget, and hence achieve a cost saving per unit. Again, there is a need for a profile quality metric to also be included, so that cost can be weighed against quality. Otherwise, a similar outcome to the minimising duration condition will occur, and the machine-learning algorithm will recommend using no product, as it results in the minimum possible cost.

### 2.2. Accessing and Controlling a PCR Thermocycler in a Real-Time Interactive Manner

Many PCR thermocyclers are designed to run as standalone instruments, in which PCR programs are loaded (or programmed on the instrument itself) and stored. For a smart PCR thermocycler system, it is required that the instrument is controllable by a computer, which has access to the algorithm that has been informed by previous training data. To achieve this, we purchased a dual-channel Chai Open qPCR instrument (www.chaibio.com; accessed on 5 February 2024). The dual-channel instrument had 16 wells and detected light at 508–532 nm and 573–597 nm using Solid-State LEDs and Photodiodes. The supported fluorophores were FAM, HEX, VIC and JOE. The Chai Open qPCR instrument was chosen, as it represents one of the only open-source pre-assembled systems available, with other open-source qPCR options being a work-in-progress, instructions to build a system (https://qninja.hisa.dev/; accessed on 5 February 2024), or published examples of ‘home-built’ systems that can be followed [15]. Although this work uses the Chai Open qPCR instrument, these same steps towards a smart PCR would apply to any open qPCR system.

A qPCR instrument was chosen (as opposed to a standard PCR thermocycler) due to its ability to provide fluorescence feedback, which we speak more about in Section 2.3. The Chai instrument was controlled via a web interface on a computer connected by USB. Through this interface, PCR programs could be created, started or stopped, and the fluorescence feedback monitored in real time during the PCR program. As well as a Graphical User Interface (GUI), there was also the ability to interact with the Chai via its Application Programming Interface (API). A web API defines a set of commands, or endpoints, that can be requested by an end-user by means of JavaScript instructions sent over the HTTP protocol [29]. The Chai API exposed endpoints to create, start and stop PCR experiments, and to retrieve the amplification data results of an experiment.

We invoked Chai API endpoints using our own JavaScript program, developed and tested using the web developmental tools of the Google Chrome browser. The Console section of the Chrome developmental tools exposed a Read, Evaluate, Print, and Loop environment [30]. This means it read the JavaScript code, entered into it, evaluated the code, printed the result, and then looped back to the first stage, allowing for convenient development and testing.

Invocation of Chai API endpoints via the JavaScript program allowed complete control over the Chai instrument and the insertion of any intermediate algorithmic steps. A limitation of the control of the Chai was that once a PCR program had been created, the parameters could not be altered on the fly (i.e., whilst the program was running). For a smart PCR instrument to operate in a way that is reactive to feedback in real time, this would require either an ability to alter PCR cycling conditions on the fly, or it could be accommodated by running a series of one- or two-cycle PCR programs, each having parameters set based on the previous cycle’s output. The basic setup of a series of short-cycle smart PCR programs would be in the following sequence of controls:(1)Set up the initial cycle(s) of the PCR program with default starting parameters, using various Chai API endpoints to modify experiments, stages and steps. Note: due to systems having a limited sensitivity to DNA concentration, the number of cycles run initially before any intervention is likely to be much larger than once the concentration has exceeded the threshold for detection.(2)Start a cycle running on the instrument using the “device/start” endpoint in the Chai API.(3)Retrieve the fluorescence amplification data at the end of the cycle using the “amplification_data” API endpoint.(4)Pass the retrieved amplification data to an external algorithm that determines the optimal conditions of the next PCR cycle(s).(5)Set up the next cycle(s) on the instrument with the algorithm-determined parameters using the Chai API, as before.(6)Continue through steps 2 to 5 until either the maximum number of cycles are run, or the early termination criteria are met.(7)End the program and write out the PCR conditions that were used for each cycle.

Depending on how the system is run, and the time taken to transmit the information between the PCR and host machines, there could be a noticeable delay between cycles. Attention would need to be given to the set-up of the described system, to minimise any such delay. However, given the speed of digital systems when compared to the heat dissipation rate of an insulated qPCR device, and the fact that delay times can be accounted for within the systems’ programmed timing, these delays would have minimal impact on system performance.

### 2.3. Carrying Out a PCR in a Way That Provides Actionable Real-Time Feedback

The combination of the dual-channel Chai Open qPCR instrument with a PC allowed fluorescence to be monitored in real time. In a fully realised smart PCR system, this real-time fluorescence could then be used by an algorithm that uses it to determine the parameters for the proceeding PCR cycle(s).

A qPCR instrument is most commonly used in quantitation, rather than endpoint PCR. Commercially available qPCR kits, such as QuantiFiler Trio (Thermo Fisher Scientific) or Investigator Quantiplex Pro (QIAGEN), use TaqMan^®^ probes [31,32]. These probes are labelled with two fluorescent dyes: a reporter dye at the 5′-end and a quencher dye at the 3′-end [33,34]. When the probe is intact (prior to polymerisation) the two dyes are in close proximity and the fluorescence of the reporter dye is suppressed, due to the fluorescent resonance energy transfer (FRET) which is occurring between the two molecules. In the denaturation and annealing stages of a qPCR program, the reporter dye fluorescence remains supressed, as the probe is still intact. However, during the extension stage, the 5′-exonuclease activity of DNA polymerase displaces the probes bound to the target sequence, and the reporter dye attached to the 5′-end is separated from the quencher dye at the 3′-end [33,34,35]. The separation means the energy transfer that supressed fluorescence when the two fluorophores were in close proximity is no longer occurring, and the reporter molecule can now fluoresce [33,34,35]. Typically, as the qPCR program progresses, and the number of copied DNA fragments increases, an amplification curve is generated based on the growing amount of fluorescence, which (at the end of the qPCR program) is then used to calculate a DNA concentration.

The dynamics of an endpoint PCR are quite different to that of a qPCR. In endpoint PCR, fluorophores are attached to primers, which become incorporated into amplicons as the PCR progresses. There is no need for real-time fluorescent feedback, and so there is no quenching of the fluorophores built into the system. The fluorescence is only important after the completion of the PCR and on being detected during capillary electrophoresis. Therefore, endpoint PCR does not necessarily lend itself to real-time fluorescence feedback. We considered two options to overcome this issue during our work:(A)For each DNA sample being amplified, run the endpoint PCR and a qPCR (in separate tubes) at the same time, and monitor the real-time feedback of both tubes.(B)Create a single reaction that allows real-time feedback from the endpoint PCR. For example, the creation of a combination reaction that is essentially a mixture of GlobalFiler (Thermo Fisher Scientific) and Quantiplex Pro (QIAGEN).

The advantage of option (A) is that each PCR amplification works as designed, and there are no issues of interaction of the endpoint PCR and qPCR constituents. There is also the advantage of the endpoint PCR fluorophores not obscuring the fluorescence from the qPCR. The disadvantage of option (A) (and hence an advantage of option (B)) is that being in two separate tubes means that the dynamics of the PCR occurring in the STR tube is not directly being measured and may differ from that in the qPCR tube. For example, the two reactions may have different tolerances to inhibition or DNA overloading, and while one shows an effect at a particular level, the other may not.

A difficulty with both options is that quantitation cycling conditions are quite different from STR amplification cycling conditions. A quantitative cycling program has 40 cycles (compared to 26–34 for STR amplification) and the cycling conditions are significantly shorter than a standard endpoint PCR program. For either option above to be viable, there must be an ability for a qPCR quantification curve to be produced and provide enough information for machine learning under standard GlobalFiler PCR conditions. If this is not the case, then a quite different variant may need to be explored, such as in a two-tube system pre-cycling the qPCR for 10–20 cycles, so that the relatively flat zone of the standard curve is passed, and then amplifying the remaining cycles under GlobalFiler conditions at the same time as the GlobalFiler reaction (and using the real-time feedback). Alternatively, the quantification reaction could be run completely in a standard manner and the amplification-curve information used to set the parameters for the GlobalFiler reaction to follow, but noting that this is no longer adjusting the PCR in real-time.

### 2.4. Creating a Metric That Represents the Quality of the Profile

Regardless of the goal given to a machine-learning algorithm, a key component for the system is to be able to measure the final DNA profile quality. As mentioned in Section 2.1, there are numerous metrics that could be used that relate to different aspects of quality. In our pilot study, we chose to work with single-source DNA profiles only. The reason for this choice was that with a single contributor the peaks could be easily assessed for balance, and the profile could easily be assessed to determine the completeness of the donor’s genotype, without the complicating factors of peak-sharing or masking. This choice simplified the metric that needed to be developed. When selecting an optimisation algorithm, there are two main choices: either maximise a scoring function or minimise a cost function. In this work, we decided to use a score function, where variations from the desired result were penalised (by the awarding of a negative score) and the goal of the system was to minimise the variation from the ideal response, i.e., obtain a score closer to 0. This means a score of 0 would represent the ideal response, with more negative scores representing a less desirable outcome. Note that this approach is mathematically identical to inverting the signs on the scoring components, calling it a cost function, and then minimising the result. We considered four key components when designing the metric: peak height, peak balance, process duration and artefact rejection.

(A)A component that deals with peak height. Ideally, peak heights in a DNA profile will be high enough so that they are not at risk of dropping out, but not so high that they start to cause significant pull-up artefacts (and, if pushed even further, not so high that they saturate the capillary electrophoresis detection system). To achieve this, we set an ideal peak height and the variation that would be tolerated around that ideal height. Practically, this was realised by calculating the probability density function of the measured peak height based on the ideal peak height and permitted standard deviation from that ideal peak height. A score was then calculated based on the log10 of the density of the exponential distribution at that point. The use of the logarithm ensured the decrease in score was not too severe, with height scores remaining higher in response to variations in peak height than if it were not utilised. Since the probability density function varies between 0 and 1, the logarithm was always negative. Furthermore, the probability density function was maximal, but not 1, at the ideal peak height, meaning that the logarithm of the probability density function was always negative. This resulted in an offset, where there was always a cost associated with any measured height, even the ideal value. This offset was removed (by adding a constant), such that the score associated with obtaining the ideal peak height was 0, with decreasing scores (larger negative values) related to the further away the measured height was from this ideal value.(B)A component that deals with peak balance. We chose a method that was intended to capture multiple manifestations of peak height variability. The metric was based on all peak heights in the profile (with missing peaks assigned a value of 0). This way, peak imbalance within a locus, between loci, or dropped-out alleles/loci would all be accounted for. The metric calculated the standard deviation of the peak heights and then divided this by the mean of the peak heights, to obtain the coefficient of variation (COV). The COV was used instead of the standard deviation to prevent the correlation between peak height and standard deviation always driving the metric to prefer lower peak heights. Unlike the first component (peak heights), there is no ideal amount of peak height variability, other than ideally having no peak height variability or imbalance. Therefore, we modelled the COV with an exponential distribution, so that the smaller the peak height variability the less the profile would be penalised. There is no specific amount by which to penalise peak height variability, except that it must be balanced with the peak height component. If the parameter of the exponential distribution was chosen to be too high, then the peak balance component would dominate the metric, and if it were too low, then the peak height component would dominate the metric (and it would become too tolerant of peak height variability). Figure 1 shows some combinations of peak height and peak imbalance that utilise these combinations of parameters. As with component A, the peak height variability component was assigned a score based on the log10 of the density of the exponential distribution at that point. Once again, an ideal value was 0, with less desirable outcomes resulting in increasingly negative scores.(C)A component that deals with process duration. The time it takes to run the entire procedure is an important consideration. Reducing the time taken to obtain a result is of benefit, as it means a more efficient process and additional throughput. However, care should be taken to not unduly emphasise duration at the expense of the quality of the resulting product. We decided to increase the cost in proportion to the duration of the process, specifically the total time of the denaturing and annealing cycles. Since this was but one factor, and it was added to the others, even a very high score (i.e., close to 0) would not be sufficient to overcome poor peak height and COV scores. However, it would be sufficient to influence the selection of a faster process that produced a result of acceptable quality.(D)A component that deals with artefacts. Machine-learning algorithms are known to achieve goals by finding solutions that have consequences not originally envisaged by humans [36]. For example, a DNA profile that achieved high scores in both component A and B could still be largely useless. Imagine that a machine-learning algorithm learned to suggest a set of PCR cycling conditions that lead to the presence of peaks at every allelic position (much like an allelic ladder). As long as those peaks were around 5000 relative fluorescence units (rfu), and all of similar height, this would receive a high score. However, the profile itself would be largely useless in the context of identifying the source of the DNA, due to being comprised almost entirely of artefacts. Therefore, it is likely that the profile metric will need to include an artefact component. The same as with point B), there is no ideal amount (or number) of artefacts other than none, and so an exponential distribution is likely to be appropriate. The variate being modelled by the exponential distribution could be the summed rfu of the artefact peaks, or the number of artefact peaks. In our initial work, artefacts were not observed (note that we do not count stutters as artefacts, as these occur in all programs, but a model could be set that took stutter ratios into account if this were desired by a user) and so no artefact component was added to the metric, but will likely be required as work continues. The exponential distribution parameter will have to be chosen as before, so that the presence of artefacts would be balanced against the other components. We note that there are potential difficulties associated with the incorporation of an artefact penalty, particularly when dealing with complex, mixed profiles, and when considering low analytical thresholds. Even when the ground truth is known, there may be additional peaks that arise as drop-in, which are legitimate (albeit unwanted) amplifications which perhaps should not be considered within a drop-in penalty. We suspect the way forward would be to penalize any peaks that do not align with expectation and to conduct multiple repeats for any PCR cycling conditions trialled, so that the effects of a sporadic amplification in one repeat would be minimized, as it was spread across replicates.

**Figure 1 genes-15-01196-f001:**
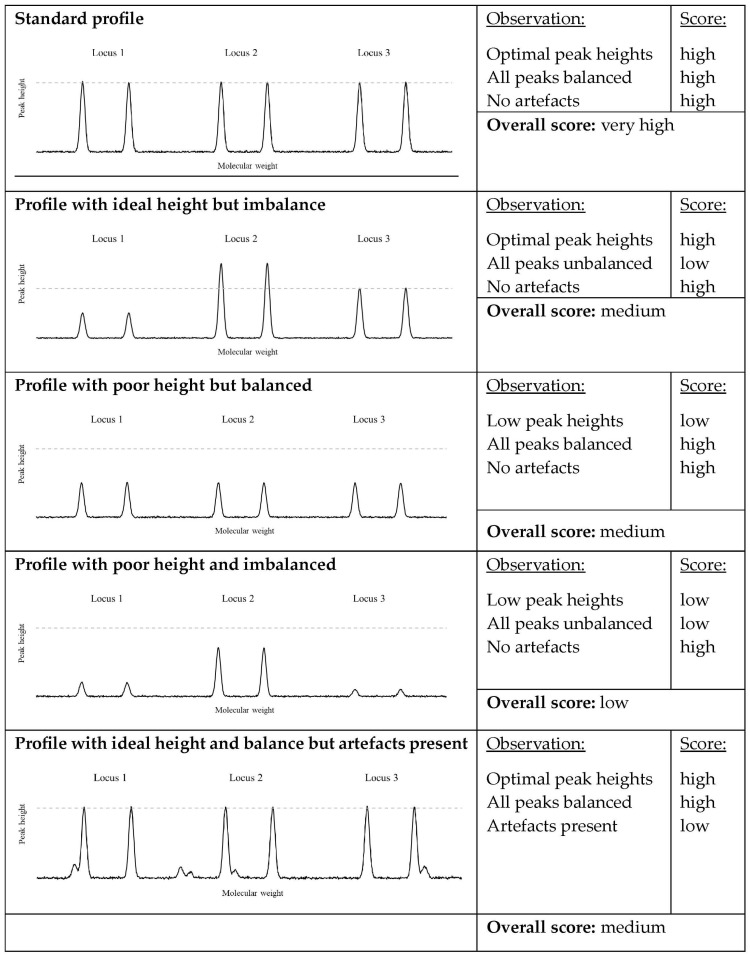
In silico constructed examples of a three-locus profile and the relative scores that would be given by the penalty distributions. The dashed line represents the optimal peak height.

The total profile score is then achieved by the weighted sum of the individual component scores.

A final point we make on the choice of metrics is the consideration of correlation. In our proposed example, it is possible that the COV will be correlated to average peak height, due to the well-known property of DNA profiles which is that as peak heights decrease, then peak height variability increases. As a demonstration of this point, we used dataset 11 from Taylor et al. [37] (see Table 1 in Reference [37]) comprising 85 single-source profiles of varying intensity, produced using the GlobalFiler™ DNA profiling kit with a 29-cycle PCR program, and visualised using a 3500xl capillary electrophoresis instrument. Figure 2 (left) shows the COV, plotted against the average peak height for alleles in the 85 profiles. There is a clear relationship of COV increasing at lower average peak heights. It may be that within a profile scoring metric, this level of correlation is acceptable, particularly considering that the ideal average peak height that the system would be designed to aim for is likely to be in the 1000s of rfu, and so above the area of greatest correlation. Alternatively, if the correlation is affecting the performance of the scoring metric, then an adjustment can be made to the COV. For example, in Figure 2 (right) we show a modified application of the COV where the average peak height is adjusted by raising to the power of 0.8 before calculation of the COV, removing the obvious trend.

### 2.5. Weighing Profile Quality against the Goal

If the goal of the machine-learning algorithm is to improve DNA profile quality, then the only score required for the algorithm to learn is the profile quality metric. If, however, the goal is something else, such as a reduction in time with minimal loss of profile quality, then there needs to be a way to weigh the goal against the profile quality score. As with the individual components of the quality score, the manner in which the goal is weighed will depend on how critical it is to achieve it. In the initial investigative phases of a machine-learning algorithm, different weights of importance could be attached to the goal, to determine an appropriate balance. We explored this concept in more detail in our pilot study.

Therefore, in our study, if DNA profile P has peaks with heights *x*_1_, …, *x*_n_, then the profile score $f\left(P\right)$ is determined by
(1)$$f\left(P\right)=K_{0}[\log_{10}\left[p\left(\bar{x}|\mu_{a},\sigma_{a}\right)\right]-C+K_{1}\log_{10}\left[p\left(c_{v}|\lambda \right)\right]+K_{2}n_{a}+K_{3}t$$
where

$\bar{x}=\frac{1}{N}\sum_{i=1}^{n}x_{i}$ is the mean of all observed peaks.$p\left(\bar{x}|\mu_{a},\sigma_{a}\right)∼N\left(\mu_{a},\sigma_{a}\right)$ is the probability density function at the point $\bar{x}$ for the distribution with mean $\mu_{a}$ (i.e., the ideal or reference mean value) and standard deviation $\sigma_{a}$ (i.e., the value for weighting variations from the ideal).${C=\log}_{10}\left[p\left(\mu_{a}|\mu_{a},\sigma_{a}\right)\right]+{log}_{10}\left(\lambda \right)$ is the constant offset to ensure an ideal peak height achieved a score of 0.$p\left(c_{v}|\lambda \right)∼\exp\left(\lambda \right)$ is the exponential distribution at the point $c_{v}$ with coeficient λ.$c_{v}=\frac{\sqrt{\frac{1}{N}\sum_{x=1}^{n}{\left(x_{i}-\bar{x}\right)}^{2}}}{\bar{x}}$ is the coefficient of variation in the peaks.$n_{a}$ is the number of observed artefact peaks.***t*** is the time (in minutes) the PCR program takes to complete.***K*** parameters weigh the importance of each element of the score. Note: while probabilities are between 0 and 1, depending on the selected value of λ, it is possible for a probability density to be greater than 1, meaning not all logarithmic responses will be less than or equal to zero. However, the inclusion of the constant **C** will ensure a maximum value of 0. This means values from both probability density scoring components will always be less negative, while the number of artefacts and the duration of the process is always positive. To ensure a more negative score indicates a response that is less desirable, all elements of the profile score function were required to be negative. Therefore, ***K*_0_** and ***K*_1_** values are positive, while ***K*_2_** and ***K*_3_** must be negative.

### 2.6. Carrying Out Machine Learning on a Set of Training Data to Inform an Algorithm for Use in the Smart PCR System

There are two types of machine learning that could be employed in projects of this type: optimisation and classification/prediction. The end goal is to produce an algorithm that can run in parallel with the PCR, take all available information, and modify the process parameters to achieve the optimal result. With linear time-invariant systems (i.e., systems that respond proportionally and have the same response to the same input at all times), the creation and control of a system model is relatively easy. The steps to do so are well-known, and the subject of numerous engineering textbooks [38]. The complexity comes with systems that do not always respond proportionally or repeatedly and which contain relatively large amounts of variability. Such systems, of which PCR is one, rely on the creation of non-linear statistical models to predict performance. These statistical models operate like a digital twin (or system state observer) of the process. They run in parallel with the real system, ingest all available data about the system’s current state, and predict the optimal process parameters to use, going forward. These digital twins operate as a classifier or predictor, and are commonly achieved with an artificial neural network (ANN). ANNs require large numbers of data to train. The reason for this is because the creation of a statistical model can only be achieved if there is information about how a process will react, on average, to a particular set of control parameters (system inputs) under a specific condition (operating point). However, once a statistical model is developed, it can be used in real time to perform closed-loop control, where the current (and possibly past) information about a process is used to estimate the best set of input (or control) parameters to produce a desired goal in the future.

Obtaining the required data to properly train an ANN capable of correctly modelling the PCR process is non-trivial, as the parameter space is enormous. Collecting data from all possible system states (conditions) to all possible input parameters is unrealistic. However, this is where the second type of machine learning, optimisation, can be employed. Optimisation is performed offline, and involves the efficient collection of a bank of training or optimisation data to determine the best possible set of parameters to apply in general. It does not require real-time knowledge of how the current process is proceeding to set the global or default parameters that should be used. The information from this open-loop (not changing parameters based on system response) optimisation procedure can then be used to train a digital twin of the process. The result of which can be used in the closed-loop (changing parameters based on system response) control of the process.

In summary, closed-loop optimisation would employ a statistical model, or digital twin, of the PCR process, in conjunction with feedback from a qPCR machine at the end of each cycle (or group of cycles). The model would then use this information to determine what alterations are required to the parameters to obtain the best possible result for a specific PCR run. The data required to train this statistical model would be collected using open-loop optimisation.

#### 2.6.1. Open-Loop Optimisation

A set of preliminary tests need to be undertaken to determine the default parameters that produce the best possible PCR outcomes. As covered in Section 2.5, once the weights for the quality (fitness) function have been determined, based on the relative importance of each component (peak height, peak balance, duration and artefact suppression), there is an objective way of quantifying the value of each PCR run. Together with the PCR cycle input parameters, this resulted a non-linear multiple-input optimisation problem, where the goal was to determine the set of input parameters that will produce the maximum possible value for the quality function. Solving this problem using brute force, by trying every possible input combination and then selecting the best value, is not only inefficient, but infeasible, with the number of experiments required. The way problems of this type are typically solved is to employ a metaheuristic algorithm, an intelligent search procedure designed to locate the optimal solution in an efficient manner [39]. An expansive review of such algorithms is beyond the scope of this work; however, some example approaches are covered below.

Genetic or evolutionary algorithms [40] involve treating the input parameters like genetic code and hybridising the parameters of high-ranking ‘parents’ to produce ‘offspring’ in a similar way to how evolution works. Genetic algorithms are a robust way to solve the problem, but typically involve many more experiments than is feasible to achieve in this situation [41,42].

A conceptually, and computationally, simpler approach is to utilise gradient descent [43]. In gradient descent, the difference (gradient) between the parameters of two sets of inputs is calculated. A scaled version of this difference, called a learning rate (≤1), is added to the highest-performing input set in the direction opposed to the lower-performing set. If the value of the new point is higher than one of the previous values, the process is repeated for the two new best values. If the new value of the new point is lower than both the previous points, the magnitude of the learning rate is reduced, the direction flipped (i.e., towards the lower-performing point rather than away from it), and a new point calculated. Multiple suggestions have been made to improve this basic approach, including modifying the learning rate [44] and adding momentum to the learning rate [45].

An alternative, but related, approach is particle swarm optimisation [46], where rather than having a very small number of candidate points (as in gradient descent, described above), a larger number of initial candidate points (particles) are generated. These particles are then moved across the input parameter space, based on their own history of values, as well as the best value determined by the entire swarm. There are several ways to process the movements of these particles, with a proposed adaptation being that of the Firefly algorithm [47]. The Firefly algorithm is biologically inspired, and states that a particle will be attracted to all other particles in proportion to the other particles’ quality value, and inversely proportional to the distance to the other particles.

In the context of discovering the optimal group of settings to use for a PCR cycle, the following methodology can be applied:(1)A learning rate is selected. This value must be less than 1 (otherwise it will be unstable) and could be fixed or decaying (where the value decreases over the number of iterations), depending on the number of iterations required. The larger the number of iterations, the more beneficial a decaying learning rate would be.(2)A group (swarm) of (semi-)random input parameters (each called a particle) are generated within the known constraints of the system.(3)PCR cycles are run with parameters specified by each of the particles.(4)The quality/score/fitness of the outcomes of each of these particles is calculated.(5)The distance between a particle and all the others is calculated in the parameter space.(6)The movement for a particular particle (i.e., the difference between the current and new testing parameters) is determined by adding together the respective fitness divided by the relative distance to each other particle in the swarm and scaling by the learning rate.(7)The new parameters for each of the particles are determined.(8)Steps (3) to (7) are then repeated, until the particles converge on a solution.

#### 2.6.2. Closed-Loop Optimisation

Following the determination of the optimal settings for a general set of PCRs, the goal of producing a smart PCR system necessitates monitoring the quantity of DNA on a per-cycle (or group of cycles) basis to ensure a particular run is following the expected trajectory. The ‘smart’ component of this approach is, therefore, the PCR system having the ability to set the parameters for the next cycle of the process. If the amount of DNA (as measured by the fluorescence at the end of a cycle) is undetectable after a defined number of cycles, the rate of DNA increase is too small, or the amount of DNA is larger than a threshold amount, then the experiment is terminated early. If the rate of DNA increase is higher than the minimum cut-off, but the total amount is still low, then the PCR machine is cycled again. In this way, samples with useable results are produced as soon as they are ready, those with potentially usable results are not terminated too soon, and those that do not contain sufficient replicating DNA do not continue to tie up the machine. This would be particularly useful in a reference DNA profiling paradigm, where the DNA is typically in abundance and of good quality. As well as saving PCR time, the early termination would avoid saturated profiles (which are often also accompanied by increased levels of artefacts) and hence avoid requiring a re-amplification of the samples and re-run of the PCR products.

In a closed-loop system, not only can the process be terminated early, but the parameters for each cycle can be adapted, based on the current conditions. The outcome of the open-loop process is a statistical model for the performance of the process, a set of rules that the system should follow to optimally approach the goal. In each cycle, the condition of the system is ascertained, and the pre-generated rules consulted to pick the parameters found to be the most conducive to progressing towards the goal, according to the preselected profile metric. For example, the algorithm compares the estimated concentration of DNA following a cycle (based on the fluorescence measured by the qPCR) with the set of open-loop data previously recorded. The algorithm then selects the next set of input parameters to be used, based on those that previously resulted in the highest rate of increase in the scoring metric under the closest previously tested condition.

## 3. Methods

Section 2.1, Section 2.2, Section 2.3, Section 2.4, Section 2.5 and Section 2.6 provide an outline for all the components theoretically required for a smart PCR system to be built. To prove the feasibility of this overall method, three simple proof-of-concept studies were conducted. The purpose of the first proof-of-concept study was to show the effects of altering PCR cycling conditions on the DNA profiles generated using the combined GlobalFiler and Investigator Quantiplex Pro kits for trace DNA samples (outlined in Section 3.3.1 and Section 4.1). The purpose of the second proof-of-concept study was to expand on part one and demonstrate the viability of priming the qPCR components before adding the GlobalFiler kit, aiming to improve the quantification curves (outlined in Section 3.3.2 and Section 4.2). The purpose of the third proof-of-concept study was to show the effects of altering cycling conditions upon the quantification curves generated using 50 ng of starting material, which was chosen for clearer observation of these effects (outlined in Section 3.3.3 and Section 4.3). 

### 3.1. Ethics Approval

Ethics approval for this project was obtained from the Low-Risk Human Research panel of the Social and Behavioural Research Ethics Committee at Flinders University, Australia (reference number 4915). All DNA samples provided by a volunteer were obtained with informed consent.

### 3.2. Control DNA

Cellular material was collected from a consenting donor, via a saliva sample. DNA was extracted using the DNA IQ^TM^ system (Promega, Madison, WI, USA) following the manufacturer’s DNA Isolation from Liquid Sample protocol with a final elution volume of 60 μL. Quantification of the DNA extract was performed using the Investigator^®^ Quantiplex Pro RGQ Kit^TM^ (QIAGEN, Clayton, VIC, Australia) and Rotor-Gene Q system (QIAGEN) using the validated manufacturer’s protocols. The quantification data were analysed using the QIAGEN Quantification Assay Data Handling Tool. This extracted DNA was used as a control for all subsequent experiments.

A second control DNA was obtained by purchasing male (M1) control DNA from QIAGEN.

### 3.3. DNA Amplification and STR Profiling

#### 3.3.1. Altering PCR Cycling Conditions for Trace Samples

Control DNA at a total input of 30 pg (6 µL of 0.005 ng/µL) was amplified using the GlobalFiler^TM^ and half-volume Investigator Quantiplex Pro combination outlined in McDonald et al. [48]. The standard GlobalFiler^TM^ PCR program followed the manufacturer’s recommended 30-cycle protocol, while the modified PCR program trialled a gradually increasing timing for the combined annealing/extension step across the 30 cycles, as is shown in Table 1 below. Any number of alterations could have been trialled, and this one was chosen arbitrarily, to trial the system. Five replicates of each PCR program were tested using a Chai Open qPCR Instrument (Chai, Palo Alto, CA, USA). The amplified DNA was then separated on a 3500 Genetic Analyser^TM^ (ThermoFisher Scientific, Waltham, MA, USA) using 8.5 μL Hi-Di Formamide, 0.5 μL 600 LIZ^®^ Size Standard (ThermoFisher Scientific) and 1 μL of the amplified PCR product.

#### 3.3.2. Priming qPCR Targets before Altering PCR Cycling Conditions to Improve DNA Amplification of Trace Samples

Control DNA at a total input of 30 pg (6 µL of 0.005 ng/µL) was also amplified using the combined GlobalFiler^TM^ and half-volume Investigator Quantiplex Pro combination outlined in McDonald et al. [48]; however, the quantification reaction was primed for 16 cycles using the standard qPCR protocol and the control DNA and the half-volume qPCR reagents only, before the GlobalFiler^TM^ reagents were added to the tube and a 30-cycle endpoint PCR was conducted. The standard qPCR program used to prime the reaction for 16 cycles followed the manufacturer’s recommended program. The GlobalFiler^TM^ standard protocol followed the manufacturer’s PCR program after the 16-cycle qPCR priming, while the modified GlobalFiler^TM^ protocol once again trialled a gradually increasing annealing/extension step after the 16-cycle priming, as shown in Table 2 below. Five replicates of each primed PCR program were also tested using a Chai Open qPCR Instrument. The amplified DNA was then separated on a 3500 Genetic Analyser^TM^ (ThermoFisher Scientific) using 8.5 μL Hi-Di Formamide, 0.5 μL 600 LIZ^®^ Size Standard (ThermoFisher Scientific) and 1 μL of the amplified PCR product.

#### 3.3.3. Altering qPCR Cycling Conditions to Improve Amplification Efficiency

Male Control DNA M1 (QIAGEN) at a total input of 50 ng (1 µL of 50 ng/µL) was amplified using a half-volume Investigator Quantiplex Pro reaction using various qPCR programs with altered annealing/extension steps, shown in Table 3. The standard qPCR program followed the manufacturer’s recommended protocol for the Investigator Quantiplex Pro RGQ kit. This was done to determine how the amplification efficiency would be altered by changing the timing of the combined annealing/extension step during the qPCR. Amplification efficiency was monitored in real-time, using the fluorescence feedback generated by the qPCR targets on the Chai Open qPCR Instrument. Five replicates of each qPCR program were tested. 

### 3.4. Data Analysis

#### 3.4.1. STR Data

STR data were analysed using GeneMapper^®^ ID-X (version 1.4). The allele calling thresholds for heterozygous peaks and homozygous peaks were set to 50 relative fluorescence units (RFU) and 150 RFU, respectively. Likelihood ratios (LRs) were calculated using the probabilistic genotyping software STRmix^TM^ v2.6.0 with the GlobalFiler^TM^ specific settings (available on request). The donor’s reference profile was compared to the profiles generated using the various PCR programs, and a likelihood ratio (LR) was generated. LR calculations considered the following propositions: 

**H_1_:** *The known donor is the source of the DNA*.

**H_2_:** *An unknown, unrelated individual is the source of the DNA*.

The sub-source LRs were calculated and reported based on the Australian Caucasian population [49], as this was the racial background of the known donor. STR profiles that contained 12 or more alleles were deemed “informative” and any profiles with fewer than 12 alleles were deemed “uninformative”, as per the requirements for upload to the Australian National Criminal Identification DNA Database (NCIDD). 

Any significant variation in the LRs between DNA profiles generated using the standard and altered PCR programs, and the primed and unprimed programs was identified using Kruskal–Wallis and post hoc Dunn’s tests, while any significant variation in the number of observed alleles, percent allele loss and average peak heights was identified using Welch’s ANOVA and post hoc Games–Howell tests. All statistical analyses were performed in R Studio [50] and *p*-values below 0.05 were considered significant.

#### 3.4.2. Quantification Data

Real-time amplification data from the experiments were analysed using MATLAB software (Version 2022A). Fluorescence value data were subtracted from the median baseline fluorescence value recorded within the first 16 cycles. A predefined logistic-curve model (tanh) was used to programmatically fit the baseline subtracted values, using the following equation:(2)$$y=\frac{A(1+\tanh\left(B\left(x-C\right)\right))}{2}$$
where

***Y*** = modelled output fluorescence;***X*** = cycle number or time;***A*** = amplitude of final fluorescence value;***B*** = gradient of the response;***C*** = midpoint of the fluorescence response (in either cycle number or time).

This equation was used to plot the baseline subtracted values against the overall run time instead of the cycle number. The five replicates of each altered protocol tested (Table 3) were then averaged to describe the general fluorescence response to the varying annealing time.

## 4. Results and Discussion

### 4.1. Altering PCR Cycling Conditions for Trace Samples—No qPCR Priming

All DNA profiles produced using the standard and modified GlobalFiler protocols were deemed informative as per the National Criminal Investigative DNA Database (NCIDD) upload requirements (see Appendix A). The DNA profiles generated using the modified GlobalFiler program had, on average, two fewer alleles and lower average peak heights than the profiles produced following the standard GlobalFiler program from the same amount of starting material (Table 4 and Table 5). However, differences between the peak heights and the number of alleles in the modified and standard GlobalFiler profiles was found to be statistically insignificant. When assigned profile quality scores as per the metric outlined in Section 2.4, the profiles produced using the standard GlobalFiler program had an average quality score of −12.84, while the profiles generated following the modified GlobalFiler program had an average score of −13.44 (Table 5; Figure 3). The behaviour of these scores reflected experienced expert human assessment of the quality of the DNA profiles. While this indicates that the modified program produced STR profiles of slightly lower overall quality, the quality variation observed was ultimately found to be statistically insignificant. This small variation in quality between the profiles generated using the standard and modified GlobalFiler protocols can be seen in Figure 4 and Figure 5. The LRs generated for all profiles produced using the standard and modified GlobalFiler protocols exceeded 10 billion in favour of H_1_ (see Appendix A). The average LR of the profiles generated using the standard GlobalFiler protocol was found to be 8.86 × 10^23^, while the profiles produced using the modified protocol were found to have a lower average LR of 3.65 × 10^20^ (Table 4). However, this slight variation was found to be statistically insignificant. 

These results support previous findings that gradual changes can be made to PCR programs without significantly decreasing the profile quality and evidentiary value of a given DNA profile [51], but also indicates that these altered programs can be used for samples containing trace amounts of DNA. The ability to produce profiles from trace amounts of DNA aligns with the findings of previous studies, where altered PCR programs, such as rapid programs [52,53], and extended protocols, such as those used for low-template samples [54,55] allowed DNA profiles to be generated from small amounts of starting material. The successful generation of informative DNA profiles using a combination of GlobalFiler and Investigator Quantiplex Pro reagents, in conjunction with altered PCR conditions, underscores the potential of our envisioned smart PCR system. 

### 4.2. Priming qPCR Targets before Altering PCR Cycling Conditions to Improve DNA Amplification of Trace Samples

The STR profiles generated using the primed reactions were significantly lower in quality than their unprimed counterparts. Both primed PCR programs trialled had at least one replicate that produced a DNA profile that was deemed uninformative as per the NCIDD upload requirements (see Appendix A). The DNA profiles produced using the standard GlobalFiler protocol primed with 16 qPCR cycles had, on average, 13 fewer alleles and approximately 34% higher average percent allele loss than the profiles produced using the unprimed standard GlobalFiler protocol (Table 4). The profiles generated using the modified GlobalFiler protocol primed with 16 qPCR cycles had, on average, 19 fewer alleles and approximately 50% higher average percent allele loss than the profiles produced using the unprimed modified GlobalFiler protocol (Table 4). A four-fold decrease in average peak height was observed in the primed standard GlobalFiler profiles when compared to the unprimed standard GlobalFiler profiles (Table 5); this reduction was found to be statistically significant (*p* = 3.85 × 10^−^^5^). Meanwhile, a seven-fold decrease in average peak height was seen in the primed modified GlobalFiler profiles when compared to the unprimed modified protocols (Table 5), and was also found to be statistically significant (*p* = 2.2 × 10^−^^2^). 

The LRs generated for profiles produced using the primed standard GlobalFiler protocol all exceeded one billion in favour of H_1_, except for one replicate, which provided strong support for H_1_ (777 in favour of H_1_) (see Appendix A). However, this small LR was expected, as the sample only contained six observable alleles and approximately 84% allele loss (Appendix A). The average LR of the profiles generated using this primed protocol was found to be 3.61 × 10^17^, which was six orders of magnitude smaller than the average LR of 8.86 × 10^23^ for the profiles produced by their unprimed counterpart (Table 4). Despite this, the variation in LR values between the primed and unprimed standard GlobalFiler protocols was found to be statistically insignificant. The profiles generated using the primed modified GlobalFiler protocol were found to have LRs ranging from very strong support for H_1_ (4.63 × 10^5^ in favour of H_1_) to exceeding 10 billion in favour of H_1_, with the exception of one replicate, which produced an LR of 6.00 × 10^−^^4^ (Appendix A). The low LR of this replicate was also expected, as the sample only had three observable alleles and approximately 92% allele loss (Appendix A). The average LR of the profiles generated using the primed modified protocol was calculated to be 3.34 × 10^16^, which was four orders of magnitude smaller than the average LR of 3.65 × 10^20^ for the profiles produced using the unprimed protocol (Table 4). However, once again, this variation was found to be statistically insignificant. 

When assigned profile quality scores, the profiles produced with 16 cycles of qPCR priming before the standard GlobalFiler protocol or modified GlobalFiler protocol produced average quality scores of −20.15 and −23.58, respectively (Table 5; Figure 3). When compared to the average quality scores assigned to the unprimed protocols in Section 4.1, both primed programs were found to produce profiles of significantly lower quality than their unprimed counterparts (Figure 3). The substantial decrease in profile quality observed through the lower average number of alleles, higher percent allele loss, and smaller likelihood ratios when both the standard and modified GlobalFiler^TM^ protocols were primed, indicates that qPCR target priming is detrimental to DNA profile quality. Additional peaks were observed in the VIC^TM^ (green) dye lane of STR profiles produced with a 16-cycle priming step, as can be seen in Figure 4. These peaks were consistently observed in the primed samples at approximately 93 bp, 152 bp and 305 bp in the VIC^TM^ dye lane and had an average peak height of approximately 2000 RFU (Figure 4). Due to their size, these additional peaks appeared at positions in the STR profile that aligned with bins and, as a result, were sometimes identified as alleles in GeneMapper-IDX. There is the possibility that these peaks could cause downstream interpretation issues, particularly in samples with multiple contributors, which are known to be difficult to interpret [54,56,57]. Due to the size of these targets, the wavelengths at which they are detected and the fact that they are only observable in the profiles produced from primed samples suggests that these are qPCR targets. 

Given that these peaks are not observable in any of the DNA profiles produced from the non-primed PCR protocols, it is suggested that the 16-cycle qPCR priming amplifies the qPCR targets sufficiently in these initial cycles, so that when the STR kit is added, and an endpoint PCR conducted, these qPCR targets are preferentially amplified over the STR targets, leading to the appearance of additional peaks in the DNA profile. This is supported by the lack of observable peaks at approximately 93 bp, 152 bp and 305 bp in the VIC^TM^ dye lane of all DNA profiles generated using protocols without the qPCR priming step (See Appendix A). The substantial height of the qPCR target peaks observed in profiles generated from a trace amount of starting material (30 pg) suggests that these additional peaks could potentially dominate profiles generated from ideal amounts of starting DNA (500 pg) to an even more detrimental degree.

It was hypothesized that conducting 16 cycles of qPCR before adding the endpoint PCR reagents could enhance the informativeness of the amplification data obtained during the STR cycles. This approach was considered because 50 ng of DNA did not reach the detection threshold of 500 RFU until approximately 16 cycles of qPCR. Thus, priming the qPCR targets for 16 cycles was expected to improve the monitoring of amplification efficiency during subsequent STR cycles for trace samples. However, the results indicate that any potential benefit from priming the qPCR reaction for 16 cycles before the endpoint PCR is outweighed by its negative impact on the overall quality of the profile. The successful dual amplification of both STR and qPCR targets without compromising profile quality, as observed in Section 4.1, suggests that conducting the entire reaction without prior qPCR priming is the optimal setup for achieving our ultimate goal of a smart PCR system.

### 4.3. Altering qPCR Cycling Conditions to Improve Amplification Efficiency

The maximum relative fluorescence produced by each of the trialled programs differed substantially, despite the same amount of starting material being used (Figure 5). The standard Investigator Quantiplex Pro program, with an annealing/extension time of 10 s, was found to produce the lowest maximum fluorescence of approximately 2900 RFU, with an annealing/extension time of 70 s (AEQ3) producing the highest maximum fluorescence of 5500 RFU (Figure 5). The amplification curves produced using annealing/extension times of 30 and 50 s (AEQ1 and AEQ2) had very similar average maximum fluorescence values, of approximately 4400 RFU and 4500 RFU, respectively (Figure 5). The curves produced with the longest annealing/extension time of 90 s (AEQ4) did not reach maximum fluorescence; however, it is expected that this would have been in the range of approximately 5300 RFU (Figure 5). 

However, when relative fluorescence was plotted against cycle number instead of total runtime, all programs trialled crossed the detection threshold of 500 RFU between 22 and 23 cycles and reached maximum fluorescence at approximately the 37-cycle mark (Figure 5), thus indicating that, despite the difference in total runtimes, each of the qPCR programs trialled produce very similar threshold values (C_t_). This suggests that the effect of altering the timing of this step after the first 16 cycles is minimal when detecting the C_t_ value and on the overall amplification efficiency. However, the substantial differences in maximum fluorescence observed between the qPCR programs trialled indicates that increasing the annealing/extension time does increase the amount of PCR target that is amplified, with an annealing/extension time of 70 s producing the greatest amount of PCR product (Figure 5). Increasing the time for annealing and extension to occur to 90 s was found to be detrimental to both the amount of DNA amplified (RFU) and the rate at which it was amplified (amplification efficiency). 

The results demonstrate that, despite variations in total runtime and maximum fluorescence levels across different qPCR programs, all tested programs consistently crossed the detection threshold within a narrow cycle range, typically between 22 and 23 cycles. This uniformity suggests minimal impact on the Ct value when the annealing/extension time is adjusted after the initial 16 cycles. However, significant differences in maximum fluorescence indicate that extending the annealing/extension time to 70 s can enhance the overall amount of PCR product generated. On the other hand, the similarity in maximum fluorescence between the qPCR programs with annealing/extension times of 30 and 50 s highlights the ability to obtain qPCR results faster, without causing a significant decrease in the amount of overall PCR product generated. These findings highlight the potential for currently used PCR protocols to be further optimised by altering the steps in non-traditional ways and optimising the value of our proposed smart PCR system.

## 5. Conclusions

We outline in this work the basic considerations for a smart PCR system that could have wide-reaching impacts on genetic laboratory work. We have demonstrated some proof-of-concept steps to overcome the first of the hurdles presented when working on the goal of producing a smart PCR system: being able to alter PCR conditions for trace samples without significantly compromising DNA profile quality; determining that the dual amplification of qPCR and STR targets is better than priming qPCR targets prior to endpoint PCR; and being able to alter cycling conditions for a qPCR and observe changes in the amplification curves produced.

Throughout this work, we were able to address the main aspects of setting up a system that would allow a PCR system to provide real-time feedback and allow a machine-learning algorithm to make on-the-fly changes to PCR conditions. We recognise that the greater project of developing a smart PCR system is still progressing, but we were able to progress it enough for a pilot study to be trialled in which PCR conditions were altered, and real-time feedback obtained (data shown in Part Two of this publication).

It may be necessary to consider a two-tube system, where a quantitative PCR and endpoint PCR are conducted in tandem, but in separate tubes. There are limitations to using a split-tube system, the main one being that there is a possibility that the fluorescence data obtained from the qPCR tube would not exactly reflect the amplification efficiency within the endpoint PCR tube. Thus, the capability of monitoring exact amplification kinetics that is afforded by the use of the combination kit would be lost.

While developing a framework that addresses the main hurdles raised in the introduction to building a smart PCR system, there are still unknowns that will need to be addressed in later studies. The optimal configuration of the reaction will need to be determined, specifically the ability for a mixed-quantification and STR-amplification reaction to still produce a DNA profile, while also still yielding useful real-time feedback. Our study suggests that there is the ability to create such a dual-purpose reaction. However, further investigation is needed into whether the real-time PCR and endpoint PCRs need to be run concurrently but separately, for the required feedback to be obtained and applied in real time.

It may be that even the two-tube system does not provide adequate real-time feedback to alter PCR programs on the fly. If this were the case, then there are ways in which information could still be used to optimise PCR programs for each sample. One such way would be to run the quantification reaction first, and then use the fluorescence curves from the quantification reaction to set up optimal endpoint PCR conditions. A second way would be to run a small number of cycles to prime the qPCR targets before attempting to measure concentration changes; however, as noted in this study, it would need to be for fewer than 16 cycles, so as to not create additional peaks in the STR profiles. Priming has the potential to reduce fluctuations due to noise at the expense of fewer updates and less responsiveness, but a balance between qPCR target and STR target amplifications would need to be found. Finally, if even these approaches proved unsuccessful, an open-loop-only system of machine learning could be used to trial PCR program changes, score profile quality and then iteratively change the program conditions. In such a system, it would not be possible to optimise the reaction per sample, but it could be used to devise more efficient, or quicker, programs for general use.

Also requiring investigation within the framework we have described, are the values of parameters within the profile scoring metric. These will need to be altered so that the various aspects of the DNA profile appearance are balanced both with each other, and also against the measure of the goal (for example, time taken for the PCR to reach completion). It is likely that different profiling systems would need some calibration specific to the kit that produced them, but at this stage we cannot say how much of an effect this would have.

With further research, the ability to monitor amplification kinetics in real time has the potential to expand our understanding of PCR dynamics beyond what would be explored using human-based investigation. This knowledge could then be used to create PCR programs that allow high-quality DNA profiles to be generated from samples that are currently deemed low quality. The potential exists for machine learning to be used to characterise and target a range of different issues commonly found within DNA profiles (i.e., low-template, inhibition or degradation), therefore ultimately increasing the number of samples processed by laboratories that provide useable DNA information.

There are many avenues that need to be explored before the ultimate goal of our proposed smart PCR system is realised; however, we have successfully started towards that goal by laying the groundwork to allow further experimentation to begin.

## Figures and Tables

**Figure 2 genes-15-01196-f002:**
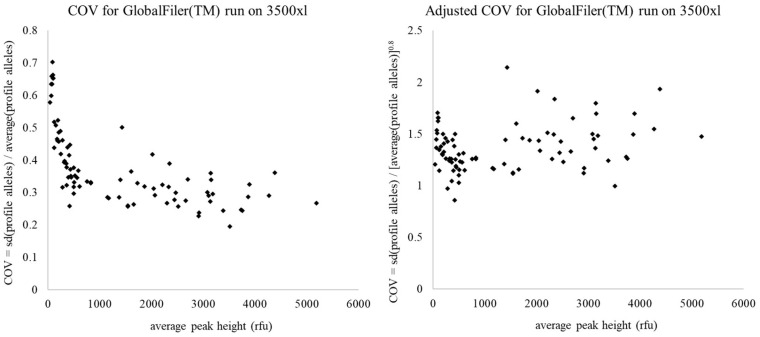
Coefficient of variation (COV) plotted against average peak height for 85 GlobalFiler™ DNA profiles produced on a 3500xl capillary electrophoresis instrument. The left shows the relationship when standard COV is calculated, and the right shows the trend when average peak heights are adjusted by raising to the power of 0.8 prior to calculation of COV. Following the power function, there is no correlation between peak height and COV.

**Figure 3 genes-15-01196-f003:**
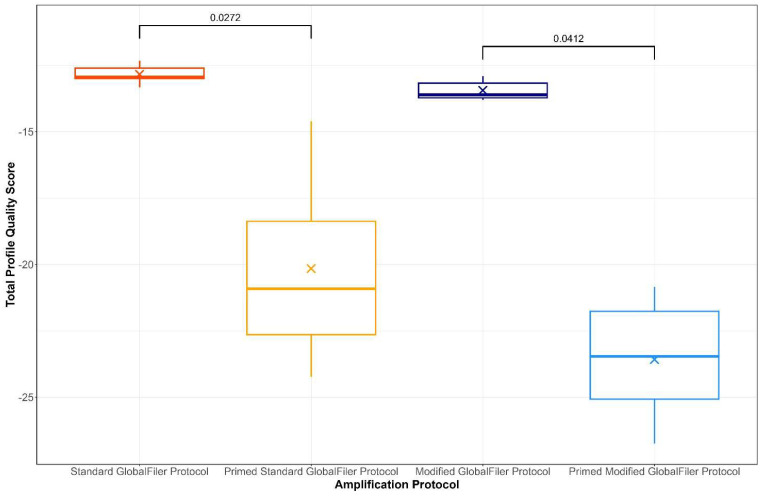
Boxplots demonstrating the spread in profile quality scores across the STR profiles produced using 30 pg of control DNA, following the standard 30-cycle GlobalFiler protocol (orange), the standard 30-cycle GlobalFiler protocol primed with 16-qPCR cycles (yellow), the modified 30-cycle GlobalFiler protocol (dark blue), and the modified 30-cycle GlobalFiler protocol primed with 16-qPCR cycles (light blue). For all PCRs, the combined GlobalFiler^TM^ and Investigator Quantiplex Pro Combination Kit was used for amplification. Significant *p*-values obtained from post hoc Games–Howell tests comparing the profile quality scores between the various PCR protocols used are shown. The means are indicated by an X, and for all protocols n = 5.

**Figure 4 genes-15-01196-f004:**
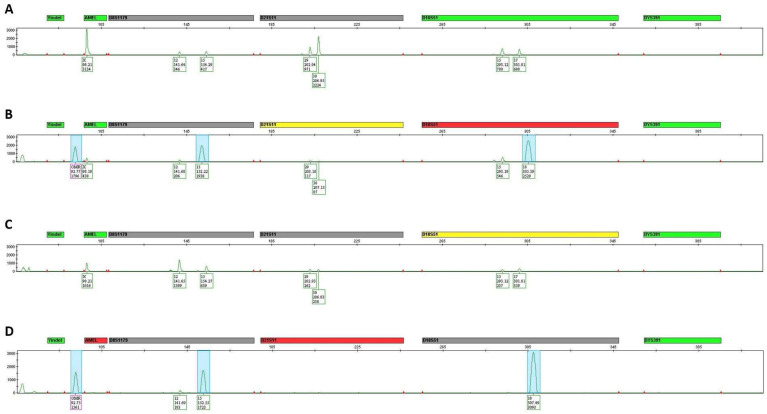
The VIC^TM^ (green) dye lane obtained from 30 pg of control DNA amplified using the standard GlobalFiler^TM^ protocol (**A**) and the 16 cycles of qPCR priming before a standard GlobalFiler^TM^ protocol (**B**), as well as the modified GlobalFiler^TM^ protocol (**C**) and the 16 cycles of qPCR priming before a modified GlobalFiler^TM^ protocol (**D**). The amplified qPCR targets that are interfering with the STR profile in this lane are highlighted in blue. Each locus label is coloured according to the quality of each locus where green indicates a pass, yellow indicates it should be checked, red indicates low quality and grey indicates that a locus has been edited.

**Figure 5 genes-15-01196-f005:**
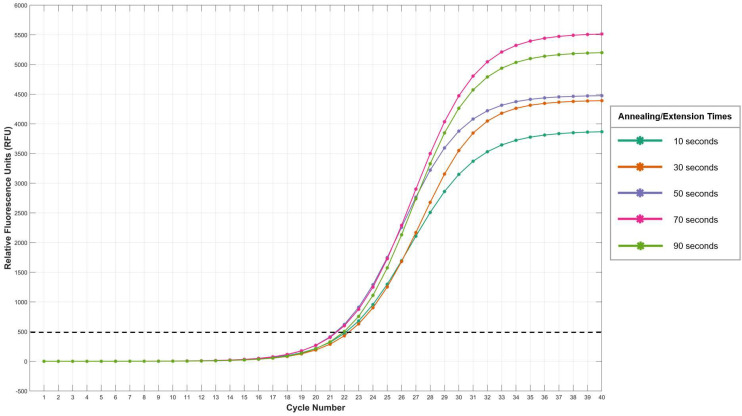
Average amplification curves generated from 50 ng of starting DNA using a standard qPCR program with altered annealing/extension steps for the last 24 cycles. The crosses indicate the averaged fluorescence at each cycle for each of the qPCR programs tested. Annealing/extension steps trialled include 10 s (Standard Investigator Quantiplex Pro Protocol—aqua), 30 s (AEQ1 Protocol—orange), 50 s (AEQ2—purple), 70 s (AEQ3—pink) and 90 s (AEQ4—green). The detection threshold (C_t_) of 500 RFU is indicated by a dashed black line. All amplification was conducted using a half-volume Investigator Quantiplex Pro reaction and fluorescence was recorded using Channel 1 (508–531 nm) on a Chai Open-Source thermal cycler. For all qPCR programs trialled, n = 5.

**Table 1 genes-15-01196-t001:** PCR programs used in part one of this study. Italicized conditions indicate variations from the standard GlobalFiler^TM^ manufacturer’s protocol.

Program	Initial Hold	Cycling Conditions (×30 Cycles)		Final Hold
		Denaturation Step	Annealing/Extension Step	
Standard GlobalFiler^TM^ manufacturer’s protocol	94 °C, 3 min	94 °C, 10 s	59 °C, 90 s	60 °C, 10 min
Modified GlobalFiler^TM^ protocol	94 °C, 3 min	94 °C, 10 s	59 °C, 60 s (+1.56 per cycle)	60 °C, 10 min

**Table 2 genes-15-01196-t002:** The primed PCR programs used in part two of this study. Italicized conditions indicate variations from the standard GlobalFiler^TM^ manufacturer’s protocol during the 30 cycles of endpoint PCR.

Program	Initial Hold	Cycling (×16 Cycles)		Initial Hold	Cycling (×30 Cycles)		Final Hold
		D	A/E		D	A/E	
Standard GlobalFiler^TM^ manufacturer’s protocol	95 °C, 3 min	94 °C, 5 s	60 °C, 10 s	94 °C, 3 min	94 °C, 10 s	59 °C, 90 s	60 °C, 10 min
Modified GlobalFiler^TM^ protocol	95 °C, 3 min	94 °C, 5 s	60 °C, 10 s	94 °C, 3 min	94 °C, 10 s	59 °C, 60 s (+1.56 s)	60 °C, 10 min

**Table 3 genes-15-01196-t003:** The qPCR programs used in part three of this study. Italicized conditions indicate variations from the standard Investigator Quantiplex Pro RGQ manufacturer’s protocol during the final 24 cycles of the total 40-cycle run.

Program	Initial Hold	Standard Cycling (×16 Cycles)		Altered Cycling (×24 Cycles)	
		D	A/E	D	A/E
Standard Investigator Quantiplex Pro RGQ manufacturer’s protocol	95 °C, 3 min	95 °C, 5 s	60 °C, 10 s	95 °C, 5 s	60 °C, 10 s
Altered annealing/extension qPCR program 1 (AEQ1)	95 °C, 3 min	95 °C, 5 s	60 °C, 10 s	95 °C, 5 s	60 °C, 30 s
Altered annealing/extension qPCR program 2 (AEQ2)	95 °C, 3 min	95 °C, 5 s	60 °C, 10 s	95 °C, 5 s	60 °C, 50 s
Altered annealing/extension qPCR program 3 (AEQ3)	95 °C, 3 min	95 °C, 5 s	60 °C, 10 s	95 °C, 5 s	60 °C, 70 s
Altered annealing/extension qPCR program 4 (AEQ4)	95 °C, 3 min	95 °C, 5 s	60 °C, 10 s	95 °C, 5 s	60 °C, 90 s

**Table 4 genes-15-01196-t004:** The average number of observed alleles, average percent allele loss and average sub-source likelihood ratios for the STR profiles produced from 30 pg of starting material using each PCR programs trialled. In brackets below, each average sub-source likelihood ratio is the range of likelihood ratios generated using that PCR program. For each program, n = 5. For all profiles, the expected number of donor alleles was 37.

PCR Program	Average Number of Observed Alleles	Average Percent Allele Loss (%)	Average Sub-Source Likelihood Ratio
Standard GlobalFiler^TM^ manufacturer’s protocol	34	9.73	8.86 × 10^23^ (9.04 × 10^17^–4.04 × 10^24^)
Primed standard GlobalFiler^TM^ manufacturer’s protocol	21	43.78	3.61 × 10^17^ (7.77 × 10^2^–1.80 × 10^18^)
Modified GlobalFiler^TM^ protocol	32	13.51	3.65 × 10^20^ (2.97 × 10^15^–1.82 × 10^21^)
Primed modified GlobalFiler^TM^ protocol	13	65.41	3.34 × 10^16^ (6.00 × 10^−4^–1.67 × 10^17^)

**Table 5 genes-15-01196-t005:** The breakdown of the average profile quality scores assigned to the STR profiles produced from 30 pg of starting material using each PCR program trialled. For each program, n = 5.

PCR Program	Average Peak Heights (RFU)			Average Penalties			Average Profile Quality Score
	Mean	Std. Dev.	COV	Peak Height	COV	Artefacts	
Standard GlobalFiler^TM^ manufacturer’s protocol	687	631	0.92	−7.44	−2.00	0	−12.84
Primed standard GlobalFiler^TM^ manufacturer’s protocol	145	287	2.05	−8.52	−4.46	−7.6	−23.58
Modified GlobalFiler^TM^ protocol	436	421	0.97	−7.93	−2.11	0	−13.44
Primed modified GlobalFiler^TM^ protocol	61	119	2.24	−8.70	−4.86	−5.2	−20.15

## Data Availability

Additional data are available in the Appendix A document.

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
