# Peer review of "Developing a Machine-Learning ‘Smart’ PCR Thermocycler, Part 1: Construction of a Theoretical Framework"

_genes, 2024, doi:10.3390/genes15091196_

Round 1
Reviewer 1 Report
Comments and Suggestions for Authors
Reviewer’s comments
Developing a Machine-Learning ‘Smart’ PCR Thermocycler Part 1: Construction of Theoretical Framework
Overall, this manuscript was well thought out and well written. The concept is novel and would be useful to the forensic community.
2. Theory
1. Line 301 – remove ‘a’ prior to STR amplification.
2. Line 307 – remove the ‘-‘ between ‘q’ and ‘PCR’ for consistency with the rest of the manuscript unless it needs to be there for some reason.
3. Line 316 – add an ‘s’ to ‘aspect’.
4. Line 317 – add a comma after ‘study’ and remove ‘work’ after study – I believe it is an extra word.
5. Line 325 – there should be a comma after i.e. – please check all instances of i.e. throughout the manuscript.
6. Line 376 – add a comma after i.e.
7. Line 404 – capitalize the ‘t’ on table.
8. Line 412 – remove one ‘then’ – there are two present.
9. Line 437 – add a comma after i.e.
10. Line 438 – add a comma after i.e.
11. Line 444 – should ‘t’ be bolded because it is a variable?
12. Line 445 – should ‘K’ be bolded because it is a variable?
13. Line 445 – should ‘weight’ be ‘weigh’?
14. Line 448 – should ‘C’ be bolded like λ is in Line 446?
15. Lines 452 & 453 – bold K0, K1, K2, K3
16. Line 459 – add comma after i.e.
17. Line 524 – add comma after i.e.
18. Line 528 – ‘decent’ should be ‘descent’.
19. Line 534 – is ‘particles’ at the end of the sentence possessive?
20. Line 535 – I think ‘proportionally’ should be ‘proportional’.
3. Methods
21. Line 593 – there is an additional space between ‘the’ and ‘Social’
22. Line 598 – Wisconsin is spelled out while California is abbreviated ‘CA’ in company citations.
23. Lines 600, 616, 617, 634, 635 – why is the ‘µ’ symbol bolded throughout this section of the manuscript?
24. Line 607 – 30 pg is an amount, not a volume. I suggest changing ‘volume’ to ‘input’ or something similar.
25. Line 612 – capitalize the ‘T’ in ‘table’.
26. Line 614 – suggest changing ‘Open-Source qPCR Thermal Cycler’ to ‘Chai Open qPCR instrument’.
27. Ine 614 – check ‘Chai Biosystems’ – it appears to be just ‘Chai’
28. Line 622 – suggest changing ‘volume’ to ‘input’ or something similar.
29. Line 631 – capitalize the ‘T’ in ‘table’.
30. Lines 632-633 – suggest changing ‘Open-Source qPCR Thermal Cycler’ to ‘Chai Open qPCR instrument’.
31. Line 633 – can remove the state and country from the in-text citation because it was in the previous paragraph.
32. Line 645 – control DNA M1 should be briefly described under 3.2 ‘Collection of Control DNA’. Potentially consider changing 3.2 sub-header to ‘Control DNA’.
33. Line 645 – ‘Qiagen’ should be in all caps ‘(QIAGEN)’.
34. Line 645 – change ‘volume’ to ‘input’ or something similar.
35. Line 647 – capitalize the ‘T’ in ‘table’.
36. Line 652 – suggest changing ‘Open-Source qPCR Thermal Cycler’ to ‘Chai Open qPCR instrument’.
37. Line 653 – can remove the state and country from the in-text citation because it was in the previous paragraph.
4. Results & Discussion
38. Line 739 – Remove ‘In’ and capitalize ‘the’. ‘In’ doesn’t make sense starting this sentence.
39. Line 744 – change the comma to a semicolon.
40. Line 905 – ‘a mixed’ and ‘amplification reactions’ – you have singular and plural here – so either remove ‘a’ before ‘mixed’ or remove the ‘s’ on ‘reactions’.
41. Line 913 – should ‘PCRs’ and ‘programs’ both be plural?
42. Line 923 – ‘…and then iterative change the program conditions’ doesn’t make sense. Should it be ‘iteratively change’?
43. Line 929 – remove ‘to’ from after ‘taken’ and place it before ‘complete’
Tables and Figures
44. Figure 2 – capitalize ‘adjusted’ above the right figure
45. Table 1 suggestion – abbreviate minutes (min) and seconds (sec) in the table and the cycling conditions will probably fit on one line and make the table appearance cleaner. It may also help to put a line through the middle so the standard and modified protocols are separated.
46. Line 618 – Table 1 caption – it looks like there is an extra space between ‘study’ and the period.
47. Line 641 – Table 2 caption – it looks like there is an extra space between ‘study’ and the period.
48. Line 654 – Table 3 caption – it looks like there is an extra space between ‘study' and the period.
49. Table 2 suggestion – abbreviate minutes (min) and seconds (sec) in the table to make the table appearance cleaner. It may also help to put a line all the way through the middle so the standard and modified protocols are completely separated. It would also help to split the table in half so the x16 and x30 cycle protocols are separate.
50. Table 3 suggestion – abbreviate minutes (min) and seconds (sec) in the table to make the table appearance cleaner. It would also help to split the table in half so the x16 and x24 cycle protocols are separate.
51. Table 4/5 – suggestion to shorten the PCR program title. In Table 5 the three modified protocols are smushed together and hard to read. Or decrease the font size so all the words fit on two lines?
52. Figure 4 – quality of this image is poor and very difficult to see.
53. Line 873 – are there crosses on the amplification curve? If so, they are not very visible. I think I see a faint circles, but I don’t think a reader would know those are crosses. The image in blurry.
54. Line 879 – suggest changing ‘Open-Source qPCR Thermal Cycler’ to ‘Chai Open qPCR instrument’.
References
55. Line 1073 – Reference 51 appears to be incomplete.
General
56. Capitalize ‘Supplementary Table’ or ‘Supplementary Figure(s)’ throughout the manuscript.
57. Be consistent with formatting of company citations.
58. Why did the authors choose to use kits from two different companies for the qPCR + PCR? For combined experiments, different chemistries from the two kits could have had some effect on the results or in Part 2.

Review thoroughly. Commas could be used more frequently throughout to separate thoughts.
Author Response
Note from the Authors:
We are grateful to the reviewers for taking the time to read the paper and provide constructive comments, especially given its length and complexity. We address each of these comments below and note where changes have been made. These changes have also been highlighted in red in the resubmitted manuscript. We agree that these changes have improved the paper, and that it now reads more clearly than before.
Reviewer 1:
Overall, this manuscript was well thought out and well written. The concept is novel and would be useful to the forensic community.
- Theory
Comment: Line 301 – remove ‘a’ prior to STR amplification.
Response: Amended
Comment: Line 307 – remove the ‘-‘ between ‘q’ and ‘PCR’ for consistency with the rest of the manuscript unless it needs to be there for some reason.
Response: Amended
Comment: Line 316 – add an ‘s’ to ‘aspect’.
Response: Amended
Comment: Line 317 – add a comma after ‘study’ and remove ‘work’ after study – I believe it is an extra word.
Response: Amended
Comment: Line 325 – there should be a comma after i.e. – please check all instances of i.e. throughout the manuscript.
Response: Thank you for highlighting this, it has now been amended in all instances across the manuscript (Lines 325, 376, 437, 438, 459 and 524).
Comment: Line 376 – add a comma after i.e.
Response: Amended
Comment: Line 404 – capitalize the ‘t’ on table.
Response: Amended
Comment: Line 412 – remove one ‘then’ – there are two present.
Response: I believe this was line 415 which has now been amended.
Comment: Line 444 – should ‘t’ be bolded because it is a variable?
Response: Amended.
Comment: Line 445 – should ‘K’ be bolded because it is a variable?
Response: Amended.
Comment: Line 445 – should ‘weight’ be ‘weigh’?
Response: Amended.
Comment: Line 448 – should ‘C’ be bolded like λ is in Line 446?
Response: Amended.
Comment: Lines 452 & 453 – bold K0, K1, K2, K3
Response: All have been amended.
Comment: Line 528 – ‘decent’ should be ‘descent’.
Response: This is now line 531 and has been amended.
Comment: Line 534 – is ‘particles’ at the end of the sentence possessive?
Response: This is now line 538 and has been amended.
Comment: Line 535 – I think ‘proportionally’ should be ‘proportional’.
Response: This is now line 538 and has been amended accordingly.
- Methods
Comment: Line 593 – there is an additional space between ‘the’ and ‘Social’
Response: Amended.
Comment: Line 598 – Wisconsin is spelled out while California is abbreviated ‘CA’ in company citations.
Response: Amended.
Comment: Lines 600, 616, 617, 634, 635 – why is the ‘µ’ symbol bolded throughout this section of the manuscript?
Response: We are not sure why this happened but thank you for highlighting it. We have amended accordingly.
Comment: Line 607 – 30 pg is an amount, not a volume. I suggest changing ‘volume’ to ‘input’ or something similar.
Response: This has been amended for clarity.
Comment: Line 612 – capitalize the ‘T’ in ‘table’.
Response: This is now line 617 and has been amended.
Comment: Line 614 – suggest changing ‘Open-Source qPCR Thermal Cycler’ to ‘Chai Open qPCR instrument’.
Response: Thank you for highlighting this, it has now been amended in all instances across the manuscript (Lines 614, 632-622, 652 and 879).
Comment: Line 614 – check ‘Chai Biosystems’ – it appears to be just ‘Chai’
Response: Amended.
Comment: Line 622 – suggest changing ‘volume’ to ‘input’ or something similar.
Response: Amended.
Comment: Line 631 – capitalize the ‘T’ in ‘table’.
Response: Amended.
Comment: Line 633 – can remove the state and country from the in-text citation because it was in the previous paragraph.
Response: This has now been removed.
Comment: Line 645 – control DNA M1 should be briefly described under 3.2 ‘Collection of Control DNA’. Potentially consider changing 3.2 sub-header to ‘Control DNA’.
Response: We have added information and changed header as suggested.
Comment: Line 645 – ‘Qiagen’ should be in all caps ‘(QIAGEN)’.
Response: Amended.
Comment: Line 645 – change ‘volume’ to ‘input’ or something similar.
Response: Amended.
Comment: Line 647 – capitalize the ‘T’ in ‘table’.
Response: Amended.
Comment: Line 653– can remove the state and country from the in-text citation because it was in the previous paragraph.
Response: Amended.
- Results & Discussion
Comment: Line 739 – Remove ‘In’ and capitalize ‘the’. ‘In’ doesn’t make sense starting this sentence.
Response: Amended.
Comment: Line 744 – change the comma to a semicolon.
Response: I believe this is referring to line 747 which has now been amended accordingly.
Comment: Line 905 – ‘a mixed’ and ‘amplification reactions’ – you have singular and plural here – so either remove ‘a’ before ‘mixed’ or remove the ‘s’ on ‘reactions’.
Response: This has been amended now.
Comment: Line 913 – should ‘PCRs’ and ‘programs’ both be plural?
Response: Amended.
Comment: Line 923 – ‘…and then iterative change the program conditions’ doesn’t make sense. Should it be ‘iteratively change’?
Response: Amended.
Comment: Line 929 – remove ‘to’ from after ‘taken’ and place it before ‘complete’
Response: Amended.
Tables and Figures
Comment: Figure 2 – capitalize ‘adjusted’ above the right figure
Response: Amended
Comment: Table 1 suggestion – abbreviate minutes (min) and seconds (sec) in the table and the cycling conditions will probably fit on one line and make the table appearance cleaner. It may also help to put a line through the middle, so the standard and modified protocols are separated.
Response: We agree that these suggestions help improve the clarity of the table. It has been updated accordingly.
Comment: Line 618 – Table 1 caption – it looks like there is an extra space between ‘study’ and the period.
Response: There is no extra space between these two words, we believe this perhaps has to do with the font.
Comment: Line 641 – Table 2 caption – it looks like there is an extra space between ‘study’ and the period.
Response: There is no extra space between these two words, we believe this perhaps has to do with the font.
Comment: Line 654 – Table 3 caption – it looks like there is an extra space between ‘study' and the period.
Response: There is no extra space between these two words, we believe this perhaps has to do with the font.
Comment: Table 2 suggestion – abbreviate minutes (min) and seconds (sec) in the table to make the table appearance cleaner. It may also help to put a line all the way through the middle so the standard and modified protocols are completely separated. It would also help to split the table in half so the x16 and x30 cycle protocols are separate.
Response: We agree that these suggestions help improve the clarity of the table. It has been edited according to the suggestions.
Comment: Table 3 suggestion – abbreviate minutes (min) and seconds (sec) in the table to make the table appearance cleaner. It would also help to split the table in half so the x16 and x24 cycle protocols are separate.
Response: We agree that these suggestions help improve the clarity of the table. It has been edited accordingly.
Comment: Table 4/5 – suggestion to shorten the PCR program title. In Table 5 the three modified protocols are smushed together and hard to read. Or decrease the font size so all the words fit on two lines?
Response: Changes have been made to the layout of tables 4 and 5 to make it easier to read.
Comment: Figure 4 – quality of this image is poor and very difficult to see. –
Response: This Figure appears clear on our version and may be an issue with the upload. We will check with the editor to ensure it is clear in final production.
Comment: Line 873 – are there crosses on the amplification curve? If so, they are not very visible. I think I see a faint circles, but I don’t think a reader would know those are crosses. The image in blurry. –
Response: Again, this Figure appears clear on our version, and we will check with the editor to ensure it is clear in final production. We would like to retain the points as we do not want to give the impression that the data has come out perfectly aligning with the fitted curve.
References
Comment: Line 1073 – Reference 51 appears to be incomplete.
Response: Thank you for bringing this to our attention, it has now been updated appropriately.
General
Comment: Capitalize ‘Supplementary Table’ or ‘Supplementary Figure(s)’ throughout the manuscript.
Response: Thank you for highlighting this, we have now amended in all instances.
Comment: Be consistent with formatting of company citations.
Response: Company citations have now been checked and updated for consistency.
Comment: Why did the authors choose to use kits from two different companies for the qPCR + PCR?
Response: We chose to use kits from different companies for the qPCR and PCR based on the instruments that we have access to and that are available in our laboratory. While we do have a range of traditional STR PCR instruments (PCR machines and CEs) we only have access to one qPCR instrument – the QIAGEN Rotor Gene-Q, and as such chose to work with kits that we can also use on this system.
Comment: For combined experiments, different chemistries from the two kits could have had some effect on the results or in Part 2.
Response: We agree with the reviewer that combining the two kits may influence the results, and this is discussed in more detail in Part 2.
Reviewer 2 Report
Comments and Suggestions for Authors
Review comments for “Developing a Machine-Learning ‘Smart’ PCR Thermocycler Part 1: Construction of Theoretical Framework”. This paper is very well written and, considering the subject material, fairly easily understood for readers not familiar with the concepts of machine learning. My comments are relatively minor in nature and are mostly for clarification on some points.
Introduction:
The introduction is long, but very thorough. The research problems and goals are very well articulated. The reader enters the paper’s work with plenty of context. Well done.
Line 38 The first use of “deoxyribonucleic acid” should be fully used prior to the acronym, as is done for STR and PCR.
Line 85 A minor point, I assume “DIY” indicates “do it yourself” but, as with any acronym, it would help to write it out. This is especially relevant for non-native English speakers who may not be familiar with the term.
Lines 88 – 91 I suggest this sentence be broken into two, as it is a bit long. Perhaps: “A real-time PCR, also known as quantitative PCR (or qPCR), is a thermocycling instrument with reaction vessel fluorescence monitoring capability. Signal from changes in amplified DNA concentration is collected throughout the PCR program and typically fed back to a system possessing an observable output, such as a personal computer (PC), in real-time.”
Theory:
Line 146 Should this sentence read “Perhaps the most impactful goal would be to improve the quality of the DNA profile”? Or “Perhaps the most impactful goal would be to improve the quality of DNA profiles”?
Lines 180 - 183 The description of the ability to terminate a PCR run early if substantial signal has been detected is particularly salient when processing known samples of pristine quality. You may want to consider mentioning that, especially in regard to avoiding over amplification of these types of samples that could lead to signal saturation and the accompanying artefacts that may require a re-amp with less template, thus wasting time.
Line 258 The appropriate word may be “actionable” (def. = capable of being acted on) rather than “actional” (def. = relating to action or an action) in the context defined in the following paragraph.
Line 316 I believe you mean “different aspects of quality.”
Line 373 I believe you mean “not unduly emphasize duration”.
Line 379 Section D regarding artefacts is certainly relevant, but how would the algorithm be able to distinguish an artefact peak from a true allelic peak? This is usually done by using both peak height and positional data on an electropherogram, but the PCR system has no separation function, so how could the metric be assessed without a reliable way to identify which peak is an allele? Simply counting the number of signal peaks may work with known samples whose ground truth # of alleles is established, but not with unknowns. Even with knowns, how can the system designate which subset of peaks are artefacts or alleles and their corresponding peak heights, which are presumably to be used as data for altering the parameters of the future cycles? I understand that this parameter was not tested in experiments detailed in the paper but, as the topic is given consideration, it would be interesting to the reader to know what the authors are thinking along these lines.
Line 391 You state that no artefacts were observed in the initial work. Did this include stutter or was stutter not considered an artefact like -A, pull-up, raised baseline, etc.? Some clarification on the subject of stutter would be useful here as most readers will be likely to immediately come to that question.
Lines 398 – 399 Figure 1 The “Peaks balanced/unbalanced” descriptor may be slightly confusing as it is referring to the peak balance among all loci (as shown), not between sister alleles as many analysts would assume. This can be easily clarified in the legend by specifying “all peaks balanced/unbalanced” and maintains consistency with the preceding text in part B).
Line 403 I believe you mean “we used dataset 11”.
Line 404 You may want to consider specifying that the “table 1” referred to is that of reference 38, so as not to confuse it with the Table 1 in the current paper. Placing a hyperlink to “table 1” would be even better, but that may be an editorial function.
Line 412 Remove one “then” as it is used twice in a row.
Methods:
Line 607 The sentence specifies a “volume of 30 pg” however, pg is not a volume. You list the total mass, a concentration, and a volume of sample used as part of a presumed total reaction volume. This should be clarified.
Line 607 How was the value of 30 pg chosen? I have no doubt that 30 pg is indeed low template for STR analysis, but it would be useful to know how the value was arrived at. Do the authors have previous data using that value, information from the literature, or is it just a good compromise?
Line 622 Same comment as 607.
Line 645 Same comment as 607.
Line 663 Why not place the GlobalFiler specific settings in the Suppl. Materials section?
Results & Discussion:
Lines 729 - 730 The conclusion here; “enhances the information that can be gained from trace DNA samples”, needs some clarification. What, exactly, has been enhanced? Every metric in Tables 4 and 5 indicates that the Standard protocol performed better or that there was no statistical difference between the Standard and Modified (with no priming for either). There is no clear “enhancement” in the information gained. Perhaps the time of the combined run, but that is not specified, nor listed in the tables.
Lines 856 – 858 The sentence “Although increasing the time for annealing and extension to occur to 90 seconds was found to be detrimental to both the amount of DNA amplified (RFU) and the rate at which it was amplified (amplification efficiency).” feels like it is incomplete. Although …. what? Perhaps just drop the word “Although”, then it is an accurate observation of data.
Conclusion:
Line 913 I do not think the “s” on PCR needs to be present. Just “used to optimize PCR programs for each sample”.
Line 926 - 927 “are the values of parameters within the profile scoring metric”?
Line 929 This sentence is somewhat awkward. Perhaps “for example time taken for the PCR to reach completion”?

Author Response
Reviewer 2:
Introduction:
The introduction is long, but very thorough. The research problems and goals are very well articulated.
The reader enters the paper’s work with plenty of context. Well done.
Comment: Line 38 The first use of “deoxyribonucleic acid” should be fully used prior to the acronym, as is done for STR and PCR.
Response: Thank you for the reminder this has now been updated accordingly.
Comment: Line 85 A minor point, I assume “DIY” indicates “do it yourself” but, as with any acronym, it would help to write it out. This is especially relevant for non-native English speakers who may not be familiar with the term.
Response: We agree that this would clarify for the reader and as such have amended the text.
Comment: Lines 88 – 91 I suggest this sentence be broken into two, as it is a bit long. Perhaps: “A real-time PCR, also known as quantitative PCR (or qPCR), is a thermocycling instrument with reaction vessel fluorescence monitoring capability. Signal from changes in amplified DNA concentration is collected throughout the PCR program and typically fed back to a system possessing an observable output, such as a personal computer (PC), in real-time.”
Response: We appreciate the suggestion and have amended this section for clarity.
Theory:
Comment: Line 146 Should this sentence read “Perhaps the most impactful goal would be to improve the quality of the DNA profile”? Or “Perhaps the most impactful goal would be to improve the quality of DNA profiles”?
Response: This sentence has been amended for clarity.
Comment: Lines 180 - 183 The description of the ability to terminate a PCR run early if substantial signal has been detected is particularly salient when processing known samples of pristine quality. You may want to consider mentioning that, especially in regard to avoiding over amplification of these types of samples that could lead to signal saturation and the accompanying artefacts that may require a re-amp with less template, thus wasting time.
Response: Thankyou, we have added this point to the discussion on early termination.
Comment: Line 258 The appropriate word may be “actionable” (def. = capable of being acted on) rather than “actional” (def. = relating to action or an action) in the context defined in the following paragraph.
Response: Thank you for highlighting this for us. We acknowledge that the word “actionable” should have been the title of section 3.2 and was likely copied across into this format incorrectly. Our apologies and it has now been updated.
Comment: Line 316 I believe you mean “different aspects of quality.”
Response: Amended.
Comment: Line 373 I believe you mean “not unduly emphasize duration”.
Response: Amended.
Comment: Line 379 Section D regarding artefacts is certainly relevant, but how would the algorithm be able to distinguish an artefact peak from a true allelic peak? This is usually done by using both peak height and positional data on an electropherogram, but the PCR system has no separation function, so how could the metric be assessed without a reliable way to identify which peak is an allele? Simply counting the number of signal peaks may work with known samples whose ground truth # of alleles is established, but not with unknowns. Even with knowns, how can the system designate which subset of peaks are artefacts or alleles and their corresponding peak heights, which are presumably to be used as data for altering the parameters of the future cycles? I understand that this parameter was not tested in experiments detailed in the paper but, as the topic is given consideration, it would be interesting to the reader to know what the authors are thinking along these lines. –
Response: we have added some text here that talks about the difficulties of applying a peak penalty related to artefacts. We suspect that the ‘teaching’ of a machine learning algorithm will always be done on construction profiles where the genotypes of donors are known (and so any peaks that do not accord with these expected peaks are counted as artefacts). So, there should not be the issue relating to having to assign the artefactual status of a peak where its ground truth status is unknown. This doesn’t account for drop-in, which we have now acknowledged in the paper. Once the machine learning algorithm has learned how to adjust PCR cycling conditions based on the training profiles, there is no further need for the outcome of DNA profiling to be scored (until the next round of training of the algorithm of course) and so the lack of ability for the PCR system to separate peaks is not needed in its ultimate implementation.
Comment: Line 391 You state that no artefacts were observed in the initial work. Did this include stutter or was stutter not considered an artefact like -A, pull-up, raised baseline, etc.? Some clarification on the subject of stutter would be useful here as most readers will be likely to immediately come to that question. –
Response: thankyou for this pickup. We do not count stutters as artefacts as they appear on all programs and are a somewhat unavoidable occurrence. Having said that, a model could be set up where the stutter ratio was one of the features being penalised (i.e. higher stutters get a greater penalty). We have added some text to this effect in the paper.
Comment: Lines 398 – 399 Figure 1 The “Peaks balanced/unbalanced” descriptor may be slightly confusing as it is referring to the peak balance among all loci (as shown), not between sister alleles as many analysts would assume. This can be easily clarified in the legend by specifying “all peaks balanced/unbalanced” and maintains consistency with the preceding text in part B).
Response: We agree that this may be a point of confusion, and as such have amended Figure 1 accordingly.
Comment: Line 403 I believe you mean “we used dataset 11”.
Response: Amended.
Comment: Line 404 You may want to consider specifying that the “table 1” referred to is that of reference 38, so as not to confuse it with the Table 1 in the current paper. Placing a hyperlink to “table 1” would be even better, but that may be an editorial function.
Response: We agree that a hyperlink would be beneficial, but also that this is likely an editorial function. We have updated the reference in text to reflect this and hope a hyperlink may be added during the later editorial stages.
Comment: Line 412 Remove one “then” as it is used twice in a row.
Response: Amended.
Methods:
Comment: Line 607 The sentence specifies a “volume of 30 pg” however, pg is not a volume. You list the total mass, a concentration, and a volume of sample used as part of a presumed total reaction volume. This should be clarified.
Response: The wording of this section and lines 622 and 645 have all been updated to “input” to clarify for the reader and in line with the comments from Reviewer 1.
Comment: Line 607 How was the value of 30 pg chosen? I have no doubt that 30 pg is indeed low template for STR analysis, but it would be useful to know how the value was arrived at. Do the authors have previous data using that value, information from the literature, or is it just a good compromise?
Response: We chose 30pg mainly based on the experience of the authors in STR DNA profiling and knowing that this sits at the general limit of where useful STR profiles are generated.
Comment: Line 663 Why not place the GlobalFiler specific settings in the Suppl. Materials section? –
Response: we have included the STRmix settings as supplementary material now.
Results & Discussion:
Comment: Lines 729 - 730 The conclusion here; “enhances the information that can be gained from trace DNA samples”, needs some clarification. What, exactly, has been enhanced? Every metric in Tables 4 and 5 indicates that the Standard protocol performed better or that there was no statistical difference between the Standard and Modified (with no priming for either). There is no clear “enhancement” in the information gained. Perhaps the time of the combined run, but that is not specified, nor listed in the tables. –
Response: We have deleted this sentence from the paragraph.
Comment: Lines 856 – 858 The sentence “Although increasing the time for annealing and extension to occur to 90 seconds was found to be detrimental to both the amount of DNA amplified (RFU) and the rate at which it was amplified (amplification efficiency).” feels like it is incomplete. Although …. what? Perhaps just drop the word “Although”, then it is an accurate observation of data.
Response: This has been amended to ensure the data is conveyed accurately.
Conclusion:
Comment: Line 913 I do not think the “s” on PCR needs to be present. Just “used to optimize PCR programs for each sample”.
Response: Amended.
Comment: Line 926 - 927 “are the values of parameters within the profile scoring metric”?
Response: Amended.
Comment: Line 929 This sentence is somewhat awkward. Perhaps “for example time taken for the PCR to reach completion”?
Response: Amended to improve clarity.